# Structural mechanism of strand exchange by the RAD51 filament

Luay Joudeh[†], Robert E Appleby[†‡], Joseph D Maman, Luca Pellegrini*

Department of Biochemistry, University of Cambridge, Cambridge, United Kingdom

## eLife Assessment

This **landmark** study describes the structure of the human RAD51 filament with a recombination intermediate called the displacement loop (D-loop). Using cryogenic structural, biochemical, and single-molecule analyses, the authors provide **compelling** evidence on how the RAD51 filament promotes strand exchange between single-stranded and double-stranded DNAs. The findings are highly relevant to the fields of homologous recombination, DNA repair, and genome stability.

*For correspondence:
lp212@cam.ac.uk

†These authors contributed equally to this work

Present address: ‡Francis Crick Institute, London, United Kingdom

Competing interest: The authors declare that no competing interests exist.

**Abstract** Homologous recombination (HR) preserves genomic stability by repairing double-strand DNA breaks and ensuring efficient DNA replication. Central to HR is the strand-exchange reaction taking place within the three-stranded synapsis wherein a RAD51 nucleoprotein filament binds to a donor DNA. Here, we present the cryoEM structure of a displacement loop of human RAD51 that captures the synaptic state when the filament has become tightly bound to the donor DNA. The structure elucidates the mechanism of strand exchange by RAD51, including the filament engagement with the donor DNA, the strand invasion and pairing with the complementary sequence of the donor DNA, the capture of the non-complementary strand and the polarity of the strand-exchange reaction. Our findings provide fundamental mechanistic insights into the biochemical reaction of eukaryotic HR.

## Introduction

The transfer of genetic information between DNA molecules promoted by homologous recombination (HR) is essential in the repair of DNA double-strand breaks (*Kowalczykowski, 2015*; *Wright et al., 2018*) and to overcome replicative stress during chromosomal DNA synthesis (*Chakraborty et al., 2023*; *Triplett et al., 2024*). The molecular apparatus of HR is also responsible for the crossover reactions that generate genetic diversity in meiosis (*Brown and Bishop, 2014*). HR deficiency causes genetic instability that predisposes to cancer (*Triplett et al., 2024*; *Prakash et al., 2015*; *Matos-Rodrigues et al., 2021*), and its synthetic lethality with other DNA repair pathways is exploited to induce cancer cell death in therapy (*Lord and Ashworth, 2017*; *Dibitetto et al., 2024*).

In eukaryotic cells, the reaction of HR is performed by RAD51 (*Baumann et al., 1996*; *Sung, 1994*), a member of a highly conserved family of ATPases that includes bacterial RecA and archaeal RadA (*Lin et al., 2006*; *Chintapalli et al., 2013*). RAD51 polymerises on the single-stranded (ss) DNA overhangs that form upon long-range resection of DNA ends (*Cejka and Symington, 2021*). The RAD51 nucleoprotein filament invades the donor DNA duplex and searches for sequence homology within a three-stranded synaptic filament intermediate (*Renkawitz et al., 2014*; *Greene, 2016*). When homology is found, the invading strand of the filament becomes stably paired to the complementary sequence of the donor DNA, yielding a structure known as a displacement loop (D-loop) (*Shibata et al., 1979*; *Jain et al., 1995*; *Piazza et al., 2019*; *Figure 1A*), which ultimately progresses to strand exchange. Although cellular HR is controlled by a complex network of recombination mediators such as BRCA2

(*Bell et al., 2023*), the RAD51 paralogues (*Greenhough et al., 2023*; *Taylor et al., 2015*; *Rawal et al., 2023*), and FIGNL1 (*Ito et al., 2023*; *Carver et al., 2025*), RAD51 can catalyse the biochemical reaction of strand exchange independently.

The RAD51/Reca/RadA protein family has been extensively investigated to elucidate the mechanistic basis for the reaction of strand exchange. Low-resolution electron microscopy (EM) and X-ray crystallography have demonstrated conservation in the helical parameters of the nucleoprotein filament (*Ogawa et al., 1993*; *Yu et al., 2001*), the structure of the ATPase domain (*Story et al., 1992*; *Pellegrini et al., 2002*), and the protomer self-association into a filament (*Conway et al., 2004*; *Wu et al., 2004*). CryoEM analysis of the RAD51 nucleoprotein filament has revealed further conservation with RecA in the mode of interaction with DNA, such as the observation that DNA stretching results from its separation into B-form nucleotide triplets (*Short et al., 2016*; *Xu et al., 2017*; *Appleby et al., 2023a*; *Chen et al., 2008*). Furthermore, single-molecule experiments have determined the kinetic parameters of filament nucleation and growth (*Galletto et al., 2006*; *Candelli et al., 2014*), have shown that a minimum length of eight homologous nucleotides is required for the formation of stable synaptic joints (*Hsieh et al., 1992*; *Qi et al., 2015*) and that homologous pairing takes place in steps of three nucleotides (*Ragunathan et al., 2011*; *Lee et al., 2015*).

High-resolution information about the RAD51 synaptic state is still missing, limiting our mechanistic understanding of the strand-exchange reaction. In this study, we have carried out the biochemical reconstitution, cryoEM structure, and biophysical validation of a D-loop of human RAD51. The structure elucidates the key steps of strand exchange, including the mechanism of filament engagement with the donor DNA, strand invasion and pairing with the complementary strand, and capture of the exchanged strand. Comparison with the synaptic state of RecA filaments (*Yang et al., 2020*) reveals both conserved and distinct features of the strand-exchange reaction, shedding light on the molecular evolution of the biochemical reaction of HR.

## Results

EM analysis of filamentous RAD51 relies normally on helical averaging, which is unsuitable for reconstruction of an aperiodic structure such as a D-loop. We therefore sought to prepare a D-loop sample of human RAD51 that is amenable to single-particle cryoEM. We designed a 32-nucleotide (nt) poly(dT) with a central, unique 10-nt-long sequence complementary to a 50-nt donor double-stranded (ds) DNA; the complementary sequence of the donor DNA was incorporated in a 10-nt mismatch bubble to favour D-loop formation (*Figure 1—figure supplement 1A* and *Supplementary file 1*). The length of the complementary sequence was decided based on evidence that RAD51 requires a homologous tract of at least eight nucleotides for stable capture of the donor DNA (*Qi et al., 2015*). As RAD51 can bind both ss- and dsDNA, we defined experimentally appropriate stoichiometry and buffer conditions for D-loop reconstitution. Electrophoretic mobility shift assay (EMSA) with fluorescently labelled oligos showed successful D-loop formation (*Figure 1—figure supplement 1B* and *Supplementary file 1*).

We collected an initial cryoEM dataset of 2792 movies of the RAD51 D-loop sample and processed the data by single-particle analysis. 2D classification showed clear evidence of the presence of synaptic particles and yielded a D-loop reconstruction at 3.31 Å resolution (*Figure 1—figure supplement 2* and *Supplementary file 2*). To improve data homogeneity and resolution, we collected a larger dataset of 8960 movies of a D-loop sample with ss- and dsDNA that had been biotinylated at both ends and capped with monomeric streptavidin (*Figure 1—figure supplement 1C*; *Appleby et al., 2023b*), yielding a structure of the D-loop at 2.64 Å (*Figure 1—figure supplement 3* and *Supplementary file 2*). The final single-particle reconstruction based on 102,596 particles comprises a nucleoprotein filament of 9 RAD51 protomers and 26 nucleotides of ssDNA, bound to a dsDNA containing the central bubble region flanked by 15- and 16-bp-long segments of donor DNA. The cryoEM map ranges in resolution between 2.5 Å and 2.9 Å for the homologous pairing region of the D-loop and 3 Å and 6 Å for the flanking dsDNA arms (*Figure 1B*).

### D-loop structure

In the D-loop structure, RAD51 maintains the right-handed helical arrangement of protomers observed in pre- and post-synaptic filaments (*Short et al., 2016*; *Xu et al., 2017*), with the same helical twist

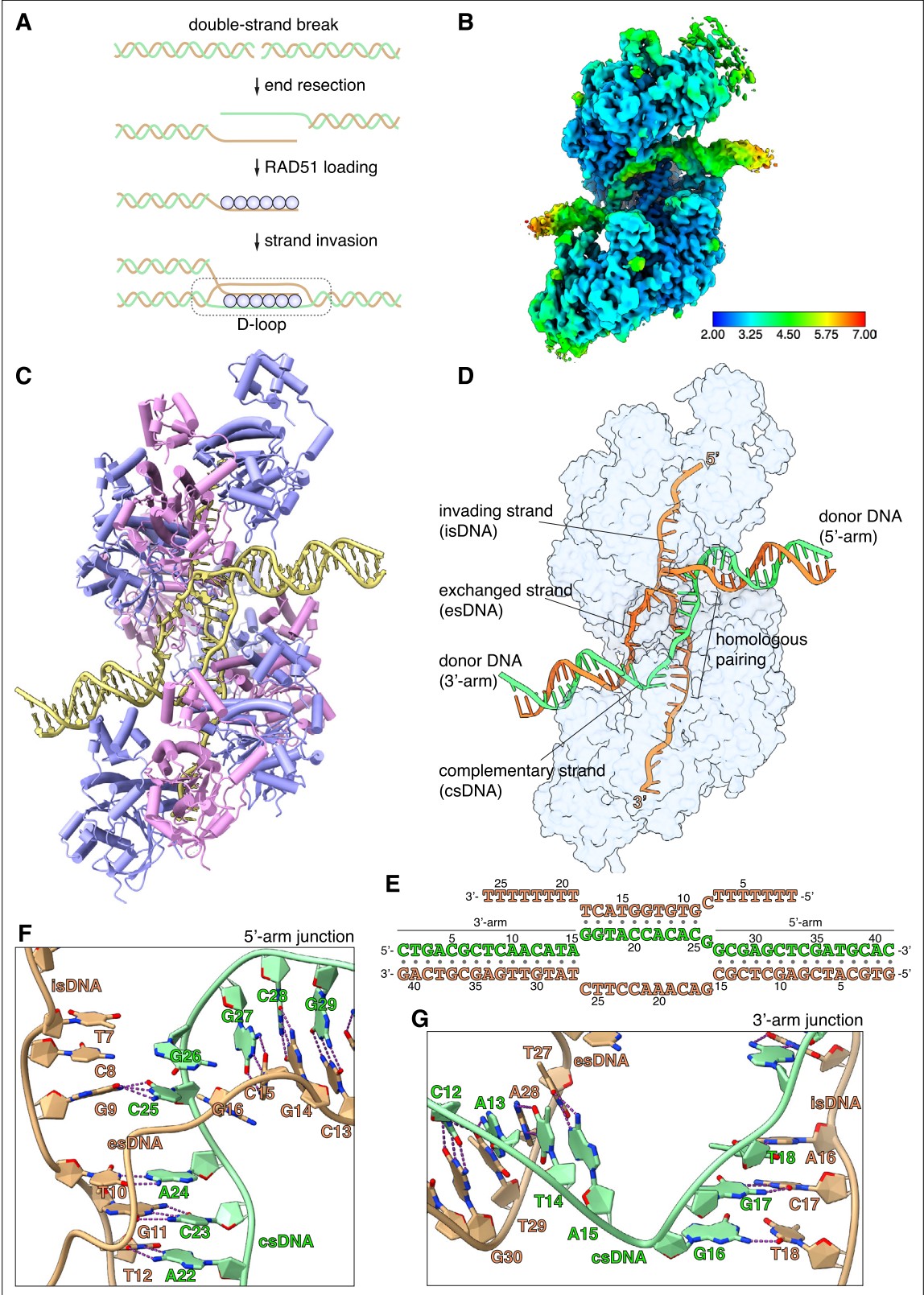

**Figure 1.** CryoEM structure of the RAD51 displacement loop (D-loop). (**A**) Schematic drawing of RAD51 D-loop formation during double-strand DNA break repair by homologous recombination. (**B**) CryoEM map of the RAD51 D-loop, coloured according to local resolution. (**C**) Drawing of the RAD51 D-loop structure. The nine RAD51 protomers in the filament are shown as cylinder cartoons, coloured alternatively in lavender and pink. The DNA strands of the D-loop are represented as light yellow tubes for the phosphoribose backbone, and the nucleotides are drawn in stick representation with

*Figure 1 continued*

filled rings. (**D**) D-loop drawing that highlights the trajectories of the DNA strands of the donor DNA and the invading DNA. The complementary and exchanged strands of the donor DNA are coloured light green and orange; the invading strand of the filament is coloured as the exchanged strand to emphasise that they would share sequence homology in vivo. The protein filament is drawn as a transparent surface in light blue. (**E**) Nucleotide sequences and observed base pairing of the D-loop DNA. Sequences are coloured as in panel **D**. (**F**) Cartoon drawing of the DNA strands at the 5'-arm junction. The phosphoribose backbone is shown as a narrow tube, the nucleotide bases as filled rings. DNA strands are coloured as in panel **D**. The Watson-Crick hydrogen bonds between base pairs are drawn as dashed purple lines. (**G**) Cartoon drawing of the DNA strands at the 3'-arm junction, coloured and annotated as in panel **F**.

The online version of this article includes the following video, source data, and figure supplement(s) for figure 1:

**Figure supplement 1.** Displacement loop (D-loop) reconstitution with human RAD51.

**Figure supplement 1—source data 1.** Original gel images for *Figure 1—figure supplement 1B and C*.

**Figure supplement 1—source data 2.** Uncropped gel images for *Figure 1—figure supplement 1A and B*.

**Figure supplement 2.** CryoEM dataset #1 (free-end DNA) processing.

**Figure supplement 3.** CryoEM dataset #2 (streptavidin-capped DNA) processing.

**Figure supplement 4.** Geometric analysis of displacement loop (D-loop) DNA.

**Figure supplement 5.** CryoEM density for homologous base pairing.

**Figure 1—video 1.** CryoEM structure of human RAD51 D-loop.

https://elifesciences.org/articles/107114/figures#fig1video1

and a slightly compressed pitch relative to the pre-synaptic filament (*Figure 1C*, *Figure 1—figure supplement 4A*, and *Figure 1—video 1*). Continuous density for the exchanged strand (es), that becomes excluded upon homologous pairing of the invading strand (is) with the complementary strand (cs) of the donor DNA, connects the arms of the donor DNA (*Figure 1—figure supplement 4B*). The filament-bound esDNA follows a peripheral path along the filament groove that runs parallel to the isDNA (*Figure 1D*). The conformation of the duplex formed by pairing of the isDNA and csDNA strands resembles closely that of the dsDNA in the post-synaptic filament (*Figure 1—figure supplement 4C*; *Xu et al., 2017*; *Appleby et al., 2023a*), with sharp bends at the junctions where the donor DNA enters and exits the filament (*Figure 1C and D*).

The arms of the donor DNA flanking the complementary sequence project away from the filament in opposite directions and with different tilt angles relative to the filament axis: the dsDNA arm closer to the 5'-end of the filament, hereinafter referred to as the 5'-arm, is almost perpendicular to the filament axis, with its filament-proximal end pointing slightly towards the 3'-end of the filament (3'-tilt), whereas the dsDNA arm closer to the 3'-end of the filament, hereinafter referred to as the 3'-arm, exhibits a more pronounced tilt in the opposite direction, towards the 5'-end of the ssDNA (5'-tilt) (*Figure 1D* and *Figure 1—figure supplement 4D*). The opposite tilt of the donor arms favours their connection via the esDNA strand in the D-loop structure.

## Homologous pairing

A continuous span of 10 base-pair interactions is present between the isDNA and csDNA (*Figure 1E* and *Figure 1—figure supplement 5A*). Base pairing includes the entirety of the csDNA sequence designed to be homologous to the invading strand, with the exception of $G26_{cs}$ (*Figure 1E and F* and *Figure 1—figure supplement 5A*): its guanine base adopts a minor-occupancy conformation in the direction of $C8$ in the isDNA but too distant for Watson-Crick hydrogen bonding, while the predominant conformation is pointing away (*Figure 1—figure supplement 5B*).

Base pairing is fully maintained in the transition between the csDNA:isDNA duplex and the flanking arms of the donor DNA (*Figure 1F and G*), with the exception of the aforementioned $G26_{cs}$. Unexpectedly, an additional, mismatched base pair is formed at the junction of the csDNA:isDNA duplex with the 3'-arm of the donor DNA, where $T18_{is}$ becomes base paired with $G16_{cs}$ (*Figure 1E, G* and *Figure 1—figure supplement 5A*). Thus, the complementary pairing observed in the D-loop structure does not entirely follow our experimental design, with one unrealised $G26_{cs}:C8_{is}$ base pair and one mismatched $G16_{cs}:T18_{is}$ base pair (*Figure 1—figure supplement 5*). This observation indicates that factors in addition to hydrogen-bonding potential contribute to determining pairing between the donor DNA and the invading strand of the filament.

## Unwinding and strand invasion of donor DNA

Biochemical and structural analyses of pre- and post-synaptic filaments have revealed that RAD51 loops L1 and L2 mediate DNA binding, with prominent roles for amino acids L1 R235 and L2 V273 in partitioning filament DNA into nucleotide triplets (*Xu et al., 2017*; *Matsuo et al., 2006*). However, L2 residues Q272 to P283 remained disordered in these filament structures (*Figure 2A*).

In our D-loop structure, the L2 sequences of four contiguous RAD51 protomers – E, F, G, and H (*Figure 2B*) – have become ordered and traceable in the density map, folding in random-coil conformation with $3_{10}$-helix elements. The structure shows that filamentous RAD51 achieves strand separation by sequential L2-loop insertion of adjacent RAD51 protomers between the strands of the donor DNA (*Figure 2C and D*). Insertion of the RAD51-H L2 loop at the 5'-arm junction causes a drastic widening of the minor groove, progressing to full separation of the complementary and exchanged strands upon L2-loop insertion by RAD51-G (*Figure 2C, D, G, and H*). In total, donor DNA unwinding by RAD51-H and -G exposes a continuous csDNA stretch of five nucleotides for homologous pairing: $A22$-$C25$ of csDNA base pair with $G9$-$T12$ of isDNA, physically organised as a base-pair triplet of $(A22$-$A24)_{cs}$ with $(T10$-$T12)_{is}$ and one unstacked $C25_{cs}$:$G9_{is}$ base pair, while $G26_{cs}$ remains unpaired (*Figures 1F, 2C, D, G, and H*).

Successive L2-loop insertions extend the unwinding of donor DNA, with each insertion exposing a triplet of csDNA nucleotides for homology pairing with the invading strand (*Figure 2C*). Thus, L2-loop insertions by RAD51-F and -E present $(A19$-$C21)_{cs}$ for base pairing with $(G13$-$T15)_{is}$, and $(G16$-$T18)_{cs}$ for base pairing with $(A16$-$T18)_{is}$, respectively (*Figure 2C–F*). After completion of homologous pairing, the strands of the donor DNA rejoin in double-helical form at the 3'-arm junction (*Figures 1G and 2C*).

The L2 loop acts as a physical spacer, directly separating the unwound strands of the donor DNA. Its sequence is dominated by hydrophobic amino acids (*Figure 2A*) that make extensive contacts with the phosphoribose backbone of the esDNA and culminate in stacking of F279's phenylalanine side chain against the last base pair of the 5'-arm duplex (RAD51-H) or the aromatic bases of nucleotides in the exchanged strand (RAD51-G, F, E) (*Figure 2E–H*). The L2 loop displays a considerable degree of conformational heterogeneity, likely because each loop engages in both common and unique contacts with the DNA. This is best exemplified by the L2 loops of RAD51-E and -H that interact with the duplex arms of the donor DNA in different ways. RAD51-H L2 initiates donor DNA unwinding by buttressing the F279 side chain against $G27_{cs}$:$C15_{is}$, the last base pair of the 5'-arm duplex (*Figure 2H*). Conversely, contact with RAD51-E L2 does not alter the conformation of the 3'-arm, as its F279 side chain stacks against C26 of the exchanged strand (*Figure 2E*).

The asymmetric binding of RAD51-H and -E L2 loops at the boundaries between the arms of the donor DNA and the three-stranded synaptic DNA suggests that strand invasion and pairing follows a preferred direction. Thus, invasion by filament DNA likely begins at the 3'-end of the homology region in the complementary strand of the donor DNA and extends in the 5'-direction, as incrementally more nucleotides of csDNA are exposed by L2-loop insertions and become available for homologous pairing.

## Capture of the exchanged strand

Strand invasion leads to displacement of the exchanged strand in the donor DNA. The structure of the D-loop shows that the esDNA is sequestered in a basic channel running parallel to the ssDNA-binding groove of the RAD51 nucleoprotein filament (*Figure 3A and B*). One side of the channel is formed by RAD51 residues 303–313 (*Figure 3C*), that fold in a beta hairpin on the inside of the filament groove, whereas the opposite side is formed by the C-terminal half of the L2-loop sequence (*Figure 3B and D*). In the filament, the beta hairpins of successive protomers align to generate a continuous ssDNA-binding stripe (*Figure 3D*). The esDNA is bound in the basic channel so that its aromatic bases remain largely exposed to solvent whereas the phosphodiester backbone faces the protein (*Figure 3B and D*).

Three adjacent RAD51 protomers – F, G, and H – engage the exchanged strand in the D-loop (*Figure 3D*). The majority of contacts comprise charged interactions between four invariant RAD51 residues R303, K304, R306, and K313 and the phosphate groups of the esDNA (*Figure 3E–G*). In addition to the electrostatic interactions, R306 is part of a GRG motif at the tip of the beta hairpin that comes into close contact with the phosphoribose backbone of the esDNA and makes stacking interactions with the aromatic bases of nucleotides $A19_{es}$ and $T25_{es}$ in RAD51-H and -G, respectively

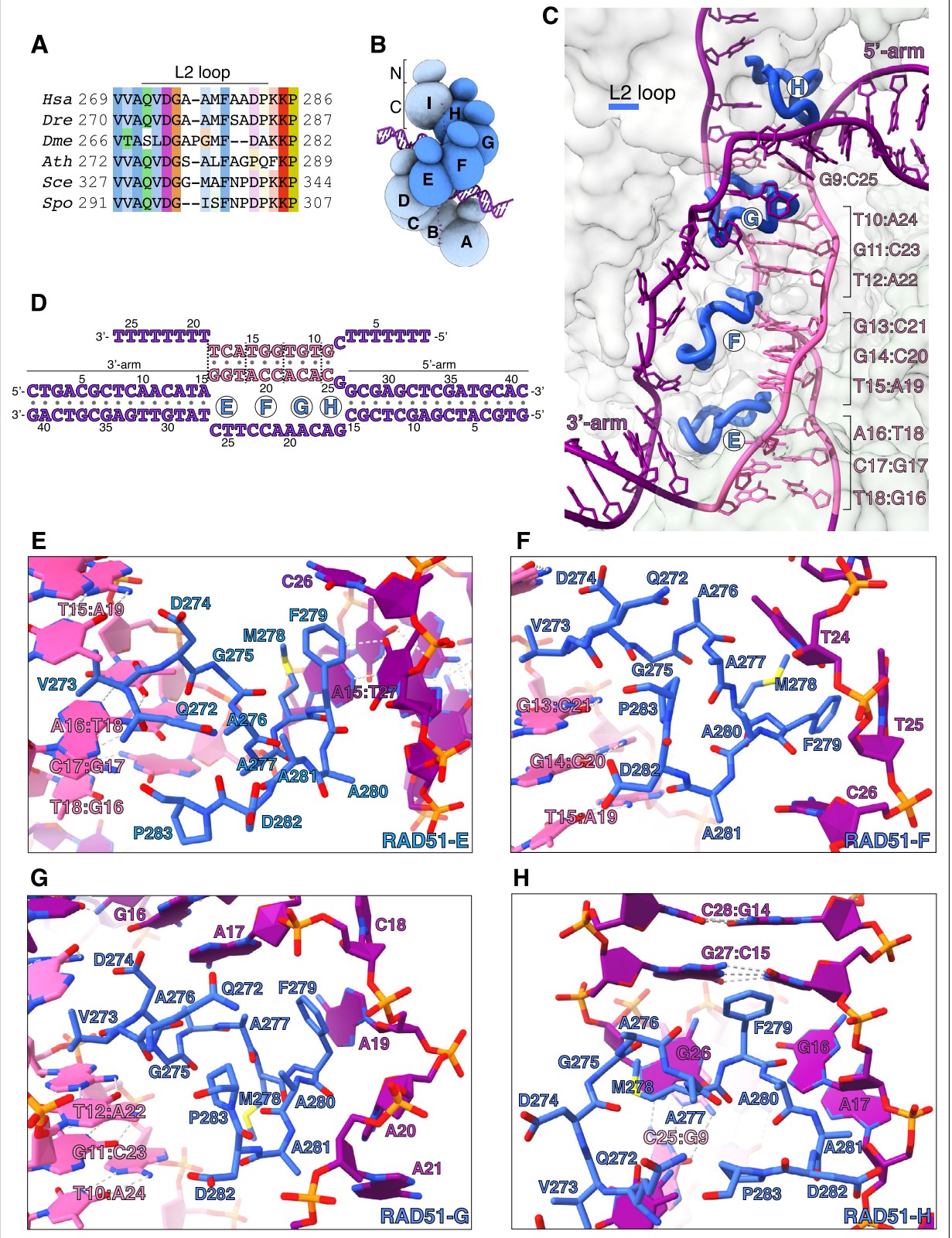

**Figure 2.** Donor DNA unwinding. (**A**) Multiple sequence alignment of RAD51 L2 loop sequences, coloured according to conservation based on the Clustal scheme (*Hsa*: human, *Dre*: zebrafish, *Dme*: fly, *Ath*: arabidopsis, *Sce*: budding yeast, *Spo*: fission yeast). (**B**) Schematic drawing of the RAD51 displacement loop (D-loop). The N- and C-terminal domains of each RAD51 protomer are shown as spheres of size proportional to mass. RAD51 E, F, G, and H are coloured in blue, the other protomers in pale blue. (**C**) Ribbon drawing of the L2 loop insertion into the donor DNA for RAD51-E, F, G, and

*Figure 2 continued on next page*

Figure 2 continued

H. The L2 loops are drawn as thick blue tubes, and the rest of the RAD51 protein is represented as transparent surface. The DNA strands are drawn as narrow tubes with the nucleotides shown as sticks; the DNA strands are coloured dark magenta except for the homologous sequences of the invading and complementary strands, coloured in pink. (**D**) Nucleotide sequences and observed base pairing of the D-loop DNA. Sequences are coloured in dark magenta and pink as in panel **C**. The approximate position of the L2-loop insertion in the unwound donor DNA is marked with the name of the RAD51 protomer. Vertical dashed lines mark the extent of donor DNA unwinding for each L2 insertion. (**E–H**) Cartoon drawings of the interface between the L2 loop of RAD51-E (panel **E**), RAD51-F (panel **F**), RAD51-G (panel **G**), RAD51-H (panel **H**) with the donor DNA. L2-loop amino acids and DNA-strand nucleotides are drawn as sticks and carbon-coloured as in panel **C**. The nucleotide bases are shown as filled rings, and the Watson-Crick hydrogen bonds are drawn as dashed lines.

---

(**Figure 3E and F**). The interaction of the 305-GRG-307 motif with esDNA resembles that of L1-loop 234-GRG-236 with csDNA in the homologous duplex of the D-loop.

Of the three RAD51 protomers F, G, and H that bind the esDNA, RAD51-G makes the most extensive set of contacts, spanning five nucleotides, from $A21_{es}$ to $T25_{es}$, involving all four basic residues R303, K304, R306, and K313 (**Figure 3F**). Binding of RAD51-G to the esDNA is further strengthened by phosphate-group contacts of L2-loop K284 with $C22_{es}$ and of R130 with $C23_{es}$ and $T24_{es}$. RAD51-H contacts four nucleotides, from $A17_{es}$ to $A20_{es}$, with R303, K304, and 305-GRG-307 (**Figure 3E**), whereas RAD51-F makes a single long-range interaction between K313 and $T25_{es}$ (**Figure 3G**). In addition, R303 and R306 of RAD51-F engage the phosphate groups of T27, A28, and T29 of the 3'-arm duplex, possibly to guide the 3'-arm in the appropriate direction as it exits the filament groove, towards the RAD51-A NTD (**Figure 3G**).

In total, the filament interacts with 9 of the 11 nucleotides of unpaired esDNA via contiguous interactions of the two RAD51 protomers G and H (5 and 4 nt, respectively). This compares with donor DNA opening by L2-loop insertion, which progresses by exposure of three nucleotides per RAD51 protomer.

## Filament binding of donor DNA arms

The D-loop structure shows that the nucleoprotein filament interacts with the double-stranded arms of the donor DNA via the N-terminal domains of RAD51 protomers A and D (**Figure 4A and B**). Although the position of the RAD51-A and -D NTDs relative to the donor DNA is broadly similar, their DNA-binding interfaces differ significantly in the extent and mode of their engagement with the 3'- and 5'-arm duplexes, respectively.

The RAD51-D NTD engages the minor groove of the 5'-arm DNA (**Figure 4C**). The side chains of K39, K40, and K64 contact the phosphate groups of 10-GCTC-13 and 34-CG-35 in the upper and lower rungs of the minor groove, respectively, while K70 and K73 contribute long-range electrostatic interactions to the binding (**Figure 4C and E**). The association of RAD51-D NTD with 5'-arm is closer and more extensive than that of RAD51-A with the 3'-arm, and includes a direct hydrogen bond between the main-chain nitrogen of G65 and the phosphate group of T12 (**Figure 4C and E**). In contrast, RAD51-A NTD binds in the major groove of the 3'-arm DNA, using electrostatic interactions mediated by the same set of lysine residues as in RAD51-D (**Figure 4D**). However, the RAD51-A association with the 3'-arm involves a more limited set of direct contacts with the phosphate groups of C5 and 31-TTG-33 across the major groove (**Figure 4D and E**).

The interaction of the NTD with the arms of the donor DNA highlights a lack of direct contacts with the bases and the predominantly electrostatic nature of the association. Such plasticity in the NTD-DNA interaction might be required for effective D-loop formation, as the direction at which the 3'-arm exits the filament upon D-loop formation is a function of the number of nucleotides involved in homologous pairing.

## Structure-based mutagenesis of RAD51 residues important for D-loop formation

The atomic model of the RAD51 D-loop has revealed several new protein-DNA interfaces, including those involved in strand separation, capture of the exchanged strand, and anchoring of the flanking DNA arms to the filament. To assess the relative importance and contribution of RAD51 residues involved in D-loop formation, we performed alanine mutagenesis of F279 (strand separation); R303, K304, R306, K313 (esDNA capture); K39, K40, K64, K70, and K73 (arm duplex binding) (**Figure 5A**).

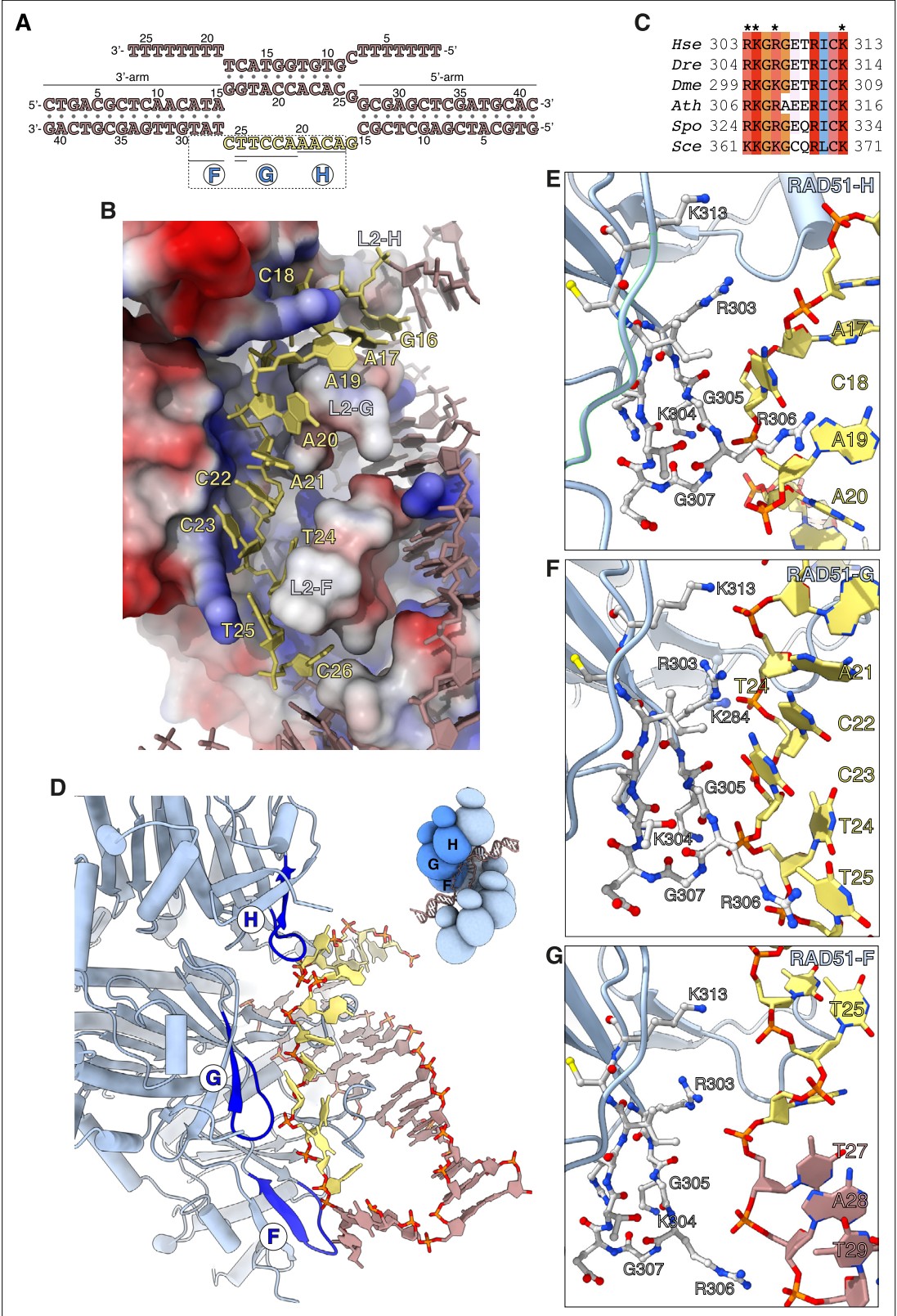

**Figure 3.** Capture of the exchanged strand. (**A**) Nucleotide sequences and observed base pairing of the displacement loop (D-loop) DNA. The exchanged strand DNA (esDNA) sequence is in yellow, the rest of the D-loop DNA in light brown. The span of esDNA bound by each of RAD51-H, -G, and -F is underlined. (**B**) Drawing of esDNA capture in a basic channel of the filament, aligned with the filament axis. The DNA is shown as a full-atom model with filled rings for the ribose and the bases, coloured in yellow for the captured esDNA nucleotides and light brown for the rest. Filament RAD51

*Figure 3 continued on next page*

*Figure 3 continued*

is shown in surface representation, coloured according to electrostatic potential from blue (basic) to red (acidic). (**C**) Multiple sequence alignment of RAD51 esDNA-binding sequences, coloured according to conservation based on the Clustal scheme (*Hsa*: human, *Dre*: zebrafish, *Dme*: fly, *Ath*: arabidopsis, *Sce*: budding yeast, *Spo*: fission yeast). (**D**) Ribbon drawing of the interface between the filament and the captured esDNA, highlighting the position of the esDNA-binding beta hairpins of RAD51-H, -G, and -F in blue. The DNA is drawn as in panel **B**. The inset shows a schematic cartoon of the RAD51 D-loop, drawn as in *Figure 2B*, to emphasise the position of RAD51-H, -G, -F in the filament. (**E–G**) Drawings of the interface between the beta hairpin amino acids of RAD51-H (panel **E**), RAD51-G (panel **F**), RAD51-F (panel **G**), and the esDNA. Beta hairpin residues and esDNA nucleotides are drawn as sticks; amino acids are coloured according to atom type, whereas esDNA nucleotides are coloured as in panel **D**.

The designed mutations are distant from the binding site of ssDNA in filamentous RAD51 and therefore are not predicted to reduce the ability of the mutant protein to form nucleoprotein filaments on ssDNA. As expected, EMSA analysis showed that the F279 and NTD mutants retained wild-type ability to form filaments on ssDNA (*Figure 5—figure supplement 1A*). However, RAD51 mutants R303A, K304A, R306A, and K313A showed a clear decrease in ssDNA binding (*Figure 5—figure supplement 1A*), suggesting that, in addition to their role in esDNA capture, they might mediate the initial contact of the incoming ssDNA during pre-synaptic filament nucleation. All RAD51 mutants showed a moderate decrease in affinity for dsDNA, with the exception of K304 (*Figure 5—figure supplement 1B*).

We evaluated the ability of the alanine mutants to perform DNA-strand exchange using an in vitro assay that measures the increase in fluorescent signal when a fluorescently labelled 60-nt oligo that is quenched in duplex form is strand-exchanged with the unlabelled sequence (*Figure 5B*; *Huang et al., 2011*). The results of the strand-exchange assay revealed that all mutants decreased the efficiency of the reaction, but with varying degrees of impairment (*Figure 5B*). The L2-loop F279A showed a modest impact, in agreement with earlier findings (*Matsuo et al., 2006*), whereas the esDNA-capture mutants R303A, K304A, R306A, and K313A displayed a more pronounced effect. Notably, the NTD mutations showed the largest negative impact on strand exchange: the double lysine-to-alanine substitutions of K39, K40 and K70, K73 reduced strand exchange to a fraction of the yield for the wild-type protein.

To provide further support for the role of the RAD51 NTD in D-loop formation, we developed a single-molecule D-loop assay using fluorescence imaging with optical tweezers (*Heller et al., 2014*). In the assay, RAD51 filaments were reconstituted on an Alexa 488-labelled 49-nt ssDNA designed to anneal at position 10.6 kb of $\lambda$ DNA (*Figure 5C*). The filaments interacted with doubly biotinylated $\lambda$ DNA held between streptavidin-coated beads and D-loop formation was monitored by the presence of a fluorescent signal at the expected locus on $\lambda$ DNA. Wild-type RAD51 filaments were capable of D-loop formation at the expected site, as assessed by kymograph and binding events analysis (*Figure 5D and E* and *Figure 5—figure supplement 2*). In contrast, filaments of the KK39,40AA RAD51 double mutant failed to show measurable binding events; a fivefold increase in the concentration of the double mutant protein showed non-specific binding and no stable D-loop formation at the expected $\lambda$ DNA site (*Figure 5E* and *Figure 5—figure supplement 2*).

These findings validate our model for the D-loop structure and highlight the important role of the RAD51 NTD in the strand-exchange reaction, by anchoring the donor dsDNA to the nucleoprotein filament in the correct orientation for the strand invasion and homology search. The biochemical behaviour of our esDNA-capture site mutants agrees with the impaired D-loop formation of a yeast RAD51 protein bearing alanine mutations at amino acids corresponding to R130, K303, and K313 of human RAD51 (*Cloud et al., 2012*).

## Discussion

The architecture of the RAD51 D-loop structure reveals the mechanism of strand invasion and homology search within a synaptic RAD51 filament. The close engagement of the RAD51-D NTD with the 5'-arm duplex, coupled to minor-groove widening by the RAD51-H L2 loop that initiates DNA unwinding, indicates that synapsis formation starts by interaction of the filament with the 5'-arm of the donor DNA. In comparison, the interaction of the RAD51-A NTD with the 3'-arm is more limited, and insertion of the L2 loop of RAD51-E does not remodel the proximal end of the 3'-arm duplex.

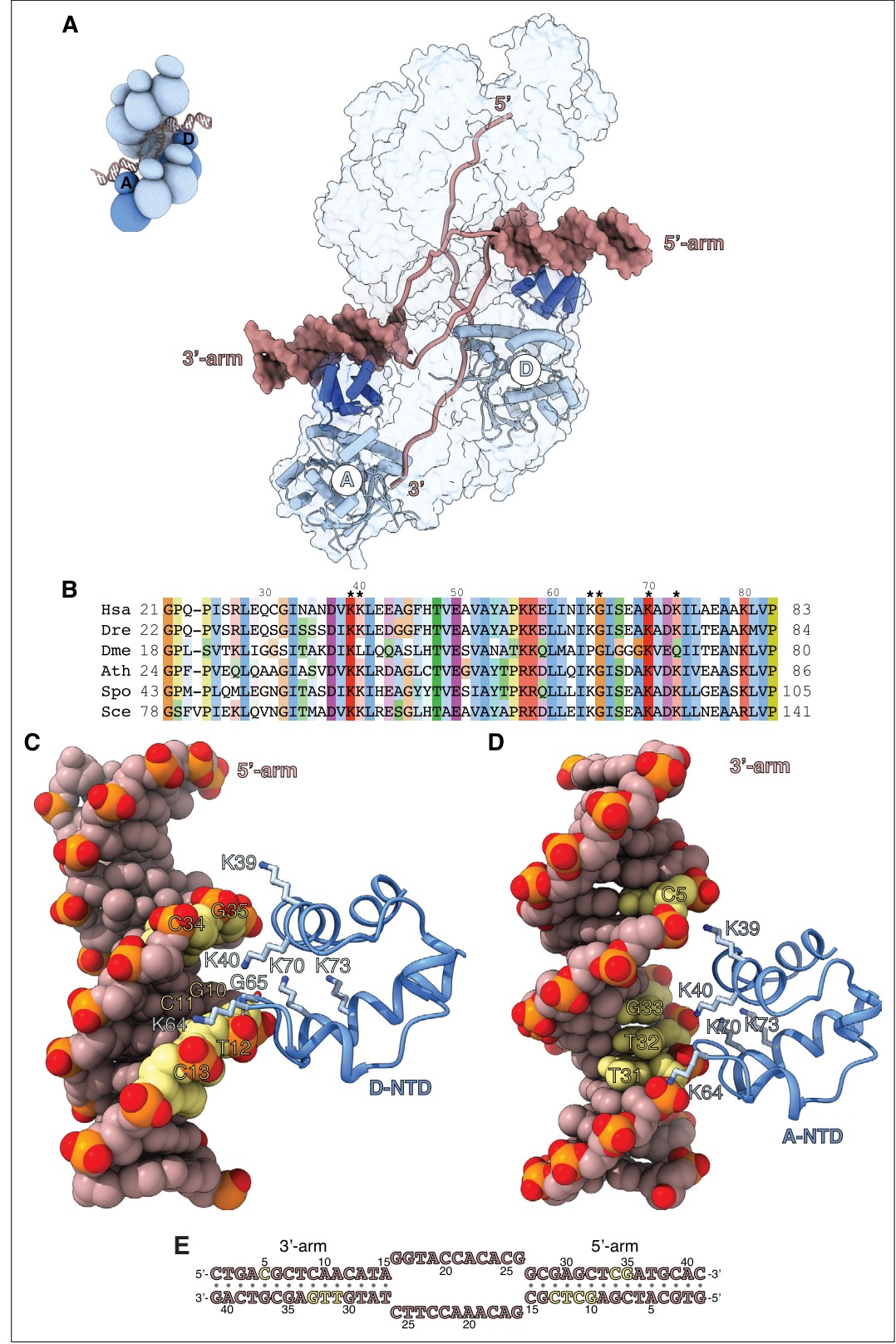

**Figure 4.** Filament engagement of the donor DNA arms. (**A**) Drawing of the displacement loop (D-loop) structure that highlights the interaction of RAD51-A and -D with the 3'- and 5'-arms of the duplex donor DNA, respectively. RAD51-A and -D are drawn as cylinder cartoons in pale blue with their NTDs in blue. The donor DNA arms are shown as molecular surfaces in light brown while the rest of the DNA is drawn as narrow tubes. The filament is

*Figure 4 continued on next page*

*Figure 4 continued*

shown as a transparent surface. The inset shows a schematic cartoon of the D-loop structure to highlight the position of the RAD51-A and -D protomers. (**B**) Multiple sequence alignment of RAD51 NTD sequences, coloured according to conservation based on the Clustal scheme (*Hsa*: human, *Dre*: zebrafish, *Dme*: fly, *Ath*: arabidopsis, *Sce*: budding yeast, *Spo*: fission yeast). (**C**, **D**) The interaction of RAD51-D and -A NTD with the 5'- and 3'-arms, respectively. The protein is drawn as a ribbon, with the side chains as sticks. The hydrogen bond between the main-chain nitrogen of G65 and the phosphate group of T12 is drawn as a dashed line. The DNA is in spacefill representation; the nucleotides contacted by the NTD are coloured yellow, and the rest of the DNA is light brown. (**E**) Nucleotide sequence of the donor DNA, coloured as in panels **C** and **D**.

---

Furthermore, the first isDNA:csDNA base pair formed at the 3'-arm junction is a T:G mismatch that is unlikely to stabilise the initial strand pairing.

Our findings form the basis for a mechanistic model of strand exchange by a RAD51 filament (*Figure 5F*). In the model, RAD51-D NTD binding directs the donor duplex towards the filament groove, where L2-loop insertions by RAD51-H and -G initiate strand separation and capture of the exchanged strand. Such unwinding exposes five nucleotides of csDNA, of which four are available for pairing with the invading strand of the filament. Initial base pair formation in the presence of homology would drive further donor DNA unwinding by L2-loop insertions of RAD51-F and -E; each insertion exposes a triplet of csDNA nucleotides, in agreement with experimental evidence that homologous pairing progresses in triplet steps (*Qi et al., 2015*; *Lee et al., 2015*). Concomitant capture of the exchanged strand by RAD51-G after insertion of a third L2 loop by RAD51-F would lead to sequestration of nine esDNA nucleotides, contributing to the formation of a stable synaptic intermediate.

Successful homology pairing drives the engagement of the 3'-arm of the donor dsDNA with the NTD of the RAD51 molecule three protomer positions away towards the 3'-end of the filament, completing D-loop formation. The observed plasticity in the interaction of the NTD with dsDNA would facilitate capture of the 3'-arm DNA emerging at slightly different angles from the filament axis, depending on the exact number of paired nucleotides. Overall, the structural features of the RAD51 D-loop provide a strong indication that strand pairing and exchange begins at the 3'-end of the complementary strand in the donor DNA and progresses with 3'-to-5' polarity (*Figure 5F*).

The observation of an unrealised base pair involving $G26_{cs}$ at the 3'-end of the unwound csDNA, which was designed to base pair with $C8_{is}$, indicates that the mechanism of unwinding of the 5'-arm prevents base pairing of the first available csDNA nucleotide, likely due to physical constraints. Further unwinding of the 5'-arm to allow $G26_{cs}$ pairing was not observed, possibly because the presence of three consecutive G:C base pairs in the 5'-arm duplex acted as a GC clamp.

The presence of a $T18_{is}:G16_{cs}$ mismatch as the last base pair at the 5'-end of the paired csDNA is intriguing (*Figure 1—figure supplement 5*). Of note, $T18_{is}$ and $G16_{cs}$ do not form a 'wobble' base pair (*Hunter et al., 1987*), indicating that fulfilment of hydrogen bonding potential is not critical. Rather, mismatch pairing that completes a base-pair triplet might occur regardless of perfect homology, to achieve a full complement of base stacking and protein-DNA interactions with RAD51 L1 and L2 loops. This observation provides a potential mechanism for how RAD51 can tolerate mismatches at certain positions of the homologous sequence during strand exchange.

The observed 3'-to-5' polarity of exchange of the complementary strand by RAD51 is opposite to the 5'-to-3' polarity of bacterial RecA (*Figure 5—figure supplement 3*), which was determined based on cryoEM structures of RecA D-loops (*Yang et al., 2020*). Comparison of RAD51 and RecA D-loop structures reveals that their different polarity follows from the opposite position occupied by their dsDNA-binding domains in the filament: in a vertically oriented filament with the 5'-end of the ssDNA at the top and the 3'-end at the bottom, the RAD51 NTDs line the bottom of the filament groove whereas the RecA CTDs hang from the top (*Figure 5—figure supplement 4*). The shared mode of minor-groove binding by RAD51-NTD and RecA-CTD and L2-loop insertion into the minor groove half a turn away leads naturally to opposite polarity in the progression of strand invasion and pairing with the complementary strand of the donor DNA. Our mechanistic interpretation of static D-loop structures awaits full reconciliation with earlier efforts to determine strand-exchange polarity for RecA and RAD51 measured under a variety of experimental conditions (*Gupta et al., 1998*; *Baumann and West, 1997*; *Sung and Robberson, 1995*; *Murayama et al., 2008*).

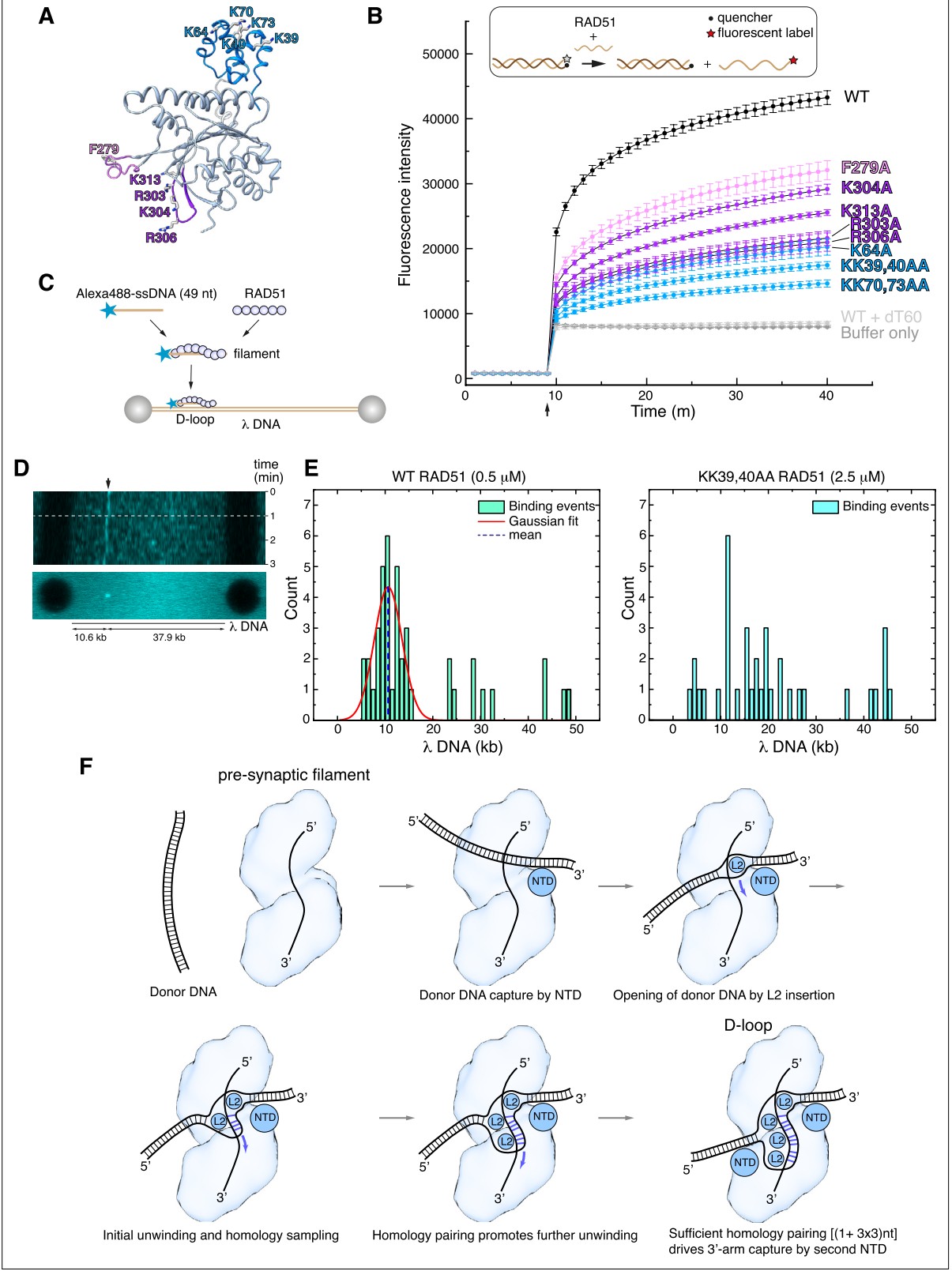

**Figure 5.** Structure-based mutagenesis of the RAD51 displacement loop (D-loop). (**A**) Ribbon drawing of RAD51, highlighting residues targeted for alanine mutagenesis, colour-coded according to their role in D-loop formation (F279 in pink; R303, K304, R306, K313 in magenta; K39, K40, K64, K70, K73 in light blue). (**B**) Strand-exchange assay showing the change in fluorescence for each RAD51 mutant following injection of the complementary strand after 9 min, with traces coloured according to amino acid as in panel **A**. Error bars denote the standard deviation of three independent replicates.

*Figure 5 continued on next page*

*Figure 5 continued*

The upper panel shows a diagrammatic representation of the assay, with DNA shown as light or dark brown lines and the fluorophore and quencher represented as in the key. (**C**) Schematic drawing of the experimental setup for single molecule analysis of RAD51 D-loop formation. (**D**) Kymograph of a representative D-loop formation event. The upper panel shows a time-lapse fluorescence intensity scan of the $\lambda$ DNA, with a stable binding event at the expected target site (10.6 kb, indicated by the arrow). The white dashed line marks the first 60 s interval that was used in frame acquisition. The lower panel shows a summed projection of the fluorescence signal over the acquisition time. (**E**) Quantification of RAD51 filament binding events. The left-side histogram displays the binding profile of wild-type RAD51 (0.5 µM), showing a Gaussian distribution of binding events centred on the expected position of the complementary sequence (~10.6 kb), consistent with specific single-stranded DNA (ssDNA) pairing and D-loop formation. The right-side histogram shows the binding profile of the KK39,40AA RAD51 double mutant (2.5 µM). (**F**) Mechanism of D-loop formation by RAD51 nucleoprotein filaments.

The online version of this article includes the following source data and figure supplement(s) for figure 5:

**Figure supplement 1.** Electrophoretic mobility shift assay for the interaction of RAD51 mutants with single-stranded DNA (ssDNA) (**A**) and double-stranded DNA (dsDNA) (**B**).

**Figure supplement 1—source data 1.** Original gel images for *Figure 5—figure supplement 1A and B*.

**Figure supplement 1—source data 2.** Uncropped gel images for *Figure 5—figure supplement 1A and B*.

**Figure supplement 2.** Fluorescence intensity traces extracted from 2D scans of trapped DNA molecules.

**Figure supplement 3.** Mechanism of displacement loop (D-loop) formation by the RecA nucleoprotein filament.

**Figure supplement 4.** Comparison of RAD51 and RecA displacement loop (D-loop) structures.

**Figure supplement 5.** Proposed steps in the functional evolution of the RecA/RAD51 recombinases.

Altogether, the available evidence suggests that the ability to perform DNA-strand exchange by members of the RecA-family of recombinases results from a process of convergent evolution (*Figure 5—figure supplement 5*). Thus, an ancestral RecA-fold ATPase protein might first have evolved to bind ssDNA using the L1 and L2 loops, possibly as a means of protecting the exposed DNA template during DNA replication. Independent acquisition of dsDNA binding by RecA and RAD51 via incorporation of carboxy- and amino-terminal domains, respectively, might have further endowed these proteins with the ability to engage simultaneously ss- and dsDNA and catalyse the exchange of homologous DNA strands.

## Materials and methods

### RAD51 expression and purification

Human RAD51 was expressed and purified as described (*Brouwer et al., 2018*). Briefly, RAD51 was co-expressed with a HisTag-MBP (Maltose Binding Protein)-BRCA2 BRC4 fusion protein in Rosetta BL21(DE3) *Escherichia coli* cells, and purified over successive steps of Ni²⁺ chromatography, heparin chromatography, and size-exclusion chromatography. The purified protein was concentrated and stored in small aliquots at –70°C. Structure-based RAD51 mutations were introduced according to standard protocols, and the mutant RAD51 proteins were purified as for the wild-type protein.

### DNA substrate preparation

DNA oligos were purchased from Integrated DNA Technologies (IDT). Unmodified or biotinylated oligos were supplied PAGE-purified. Fluorescently labelled oligos were HPLC-purified. All oligos were resuspended to 100 µM in TE buffer (10 mM Tris pH 7.5, 1 mM EDTA). dsDNA reagents were prepared at a concentration of 10 µM duplex in TE buffer, by mixing equimolar concentrations of the constituent complementary ssDNA oligos and heating at 95°C for 5 min before slowly cooling to room temperature.

For the strand-exchange assay, a duplex DNA 60mer in which one strand was 5′-labelled with fluorescein and the complementary strand 3′-labelled with an Iowa Black quencher was prepared by mixing quencher strand to fluorescent strand at a 1.2:1 molar ratio to ensure full quenching and purified on 10% native PAGE.

All DNA sequences can be found in *Supplementary file 1*.

## Electrophoretic mobility shift assay

EMSA reactions were prepared in buffer: 25 mM HEPES pH 7.5, 150 mM NaCl, 2 mM DTT, 5 mM $CaCl_2$, and 2 mM ATP, by mixing fluorescein-labelled ss- or dsDNA 60mer at 250 nM concentration with increasing concentrations of wild-type or mutant RAD51 (0.5, 1.0, 2.0, 4.0, 6.0, 8.0, and 10.0 µM) in a 20 µL volume, and incubated at room temperature for 10 min. 10 µL of 50% sucrose (wt/vol) was added to each reaction mixture prior to electrophoresis, and samples were resolved on a 0.5% agarose gel in TB buffer (45 mM Tris, 45 mM boric acid) at 45 V for 2 hr at 4°C.

For D-loop reconstitution experiments, EMSA reactions were prepared in the same buffer by mixing 500 nM of either Cy5-labelled ssDNA or doubly biotinylated ssDNA with increasing concentrations of RAD51 (1.0, 2.5, 5.0, 10.0, 20.0, 30.0 µM), followed by incubation at room temperature for 10 min. 500 nM of either Cy3-labelled dsDNA or doubly-biotinylated Cy3-labelled DNA was then added to the reaction and incubated for a further 10 min before electrophoresis. In experiments with mono-streptavidin (mSA) capped DNA, mSA was first added to the DNA at 5× molar excess over biotin labels and incubated for 15 min at room temperature before adding it to EMSA reactions. SAvPhire mSA was purchased from Sigma-Aldrich and resuspended to a working stock of 4 mg/mL from lyophilised powder according to supplier instructions.

In experiments where unlabelled DNA was used, the gel was stained in a 2× SYBR Gold solution made up in milliQ water (10 min, 25°C) and then destained in milliQ water. Gels were visualised using a Typhoon FLA 9000 fluorescent gel imager, by excitation at 473 nm for fluorescein-labelled DNA or SYBR Gold staining, 532 nm excitation for Cy3-labelled DNA or 635 nm excitation for Cy5-labelled DNA.

## DNA strand-exchange assay

Strand-exchange assays were performed using a PHERAstar FS instrument (BMG Labtech) equipped with a fluorescence intensity optic module (excitation at 485 nm, emission at 520 nm, 10 nm bandwidth). The gain was calibrated to 80% of the maximum fluorescence intensity using a 25 nM sample of fluorescein-labelled ssDNA in strand-exchange buffer: 150 mM NaCl, 25 mM HEPES pH 7.5, 2 mM DTT, 2 mM $CaCl_2$, 2 mM ATP, 0.1 mg/mL BSA.

Reactions were prepared in 50 µL volumes of strand-exchange buffer containing 500 nM RAD51 and 50 nM ssDNA 60mer, corresponding to a 1:2 ratio of RAD51 to RAD51-binding sites in the DNA. RAD51 was incubated with ssDNA at 30°C for 9 min prior to initiation of the strand-exchange reaction by 50 µL injection of dsDNA labelled with fluorescein at the 5'-end of one strand and with the Iowa Black quencher at the 3'-end of the complementary strand, using the PHERAstar FI Slow Kinetics mode. Fluorescence measurements began 2 s after duplex injection, and data were collected at 1 min intervals for an additional 30 min. As a specificity control for strand exchange, the reaction was performed with an oligo(dT) ssDNA 60mer. Blank measurements were performed by injecting dsDNA into strand-exchange buffer lacking both ssDNA and RAD51. All reactions were performed in triplicate.

## CryoEM

### Sample preparation

5 µM RAD51 was first incubated with 250 nM ssDNA in buffer: 150 mM NaCl, 25 mM HEPES pH 7.5, 2 mM DTT, 2 mM ATP, 5 mM $CaCl_2$, for 10 min to assemble a nucleoprotein filament; 250 nM dsDNA was then added to the sample and incubated for a further 10 min. In experiments with mSA-capped DNA, mSA was first added to the DNA at 5× molar excess over biotin labels and incubated for 15 min before the addition of other components.

### Vitrification protocol

UltrAuFoil R1.2/R1.3 300 mesh gold grids (Quantifoil) were glow-discharged twice for 1 min using a PELCO easiGlow system (0.38 mBar, 30 mA, negative polarity). 3 µL of sample was applied to the mesh side of each grid before plunge-freezing in liquid ethane using a Vitrobot Mark IV robot (FEI), set to 100% humidity, 4°C, 2 s blot time, and –3 blot force.

### Grid screening and data collection

Grids were screened using a Talos Arctica 200 keV transmission electron microscope fitted with a Falcon III direct electron detector (Thermo Fisher). Data were collected using a Titan Krios G3 300 keV

transmission electron microscope (Thermo Fisher) fitted with a K3 (Gatan) direct electron detector, using the EPU package (Thermo Fisher). Two datasets were collected: a first dataset with free-end DNA (#1; 2792 movies in super-resolution mode at 0.326 Å/pixel) and a second dataset with mSA-capped ss- and dsDNA (#2, 8960 movies in super-resolution mode at 0.326 Å/pixel and 2× binned). All data were recorded at the cryoEM facility of the University of Cambridge in the Department of Biochemistry. Data collection parameters are reported in *Supplementary file 2*.

## Data processing

Data was processed using cryoSPARC 4.4.1 (*Punjani et al., 2017*). Micrographs were motion-corrected with Patch Motion Correction with output F-crop factor set to ½. The contrast transfer function (CTF) was estimated using Patch CTF Estimation, and exposures were manually curated to remove outliers.

Dataset #1 1477 particles were manually picked from 2711 micrographs and, after 2D classification, 578 particles were selected to generate filament templates for automated picking. Template picking yielded 948,693 particles, which were reduced to 838,403 after outlier removal. 705,678 particles were extracted using a 400-pixel box. Multiple rounds of 2D classification led to the identification of a set of 2522 particles that yielded classes with clear evidence of synaptic filaments and was therefore used for Topaz training. The trained Topaz model picked 66,799 particles, which were manually curated to 64,942 particles; of these, 60,328 were extracted with a 416-pixel box. Ab initio reconstruction of two 3D volumes followed by heterogeneous refinement yielded a 3D shape of a recognisable D-loop structure. The corresponding set of 44,864 particles was subject to 3D classification into five classes, three of which presented pronounced D-loop features with strong density for the synaptic dsDNA; these classes were pooled and the combined 29,398 particles refined by homogenous refinement to 4.43 Å. The resulting particle set was used for a second round of Topaz training and picking, which yielded 167,005 particles and from which 153,235 particles were extracted with a 416-pixel box. The particles were used directly for ab initio reconstruction and heterogeneous refinement, generating a D-loop structure from a set of 117,041 particles. These were fractionated into 10 classes by 3D classification, and the particles for four of these classes that presented the best D-loop shapes (50,363 particles) were pooled and subject to homogenous refinement to 3.62 Å, followed by non-uniform refinement to yield a final D-loop reconstruction at 3.31 Å.

Dataset #2 The final 3D reconstruction obtained from dataset #1 processing was low-pass filtered at 8 Å and used to generate templates for particle picking. 2,940,045 particles were picked from 8834 micrographs, 2,680,398 particles were selected after manual curation, and 2,245,379 particles extracted with a 208-pixel box. After multiple rounds of 2D classification, classes from a 19,116 particle set showed clear synaptic features and were used for ab initio reconstruction and homogenous refinement of a 3D volume from 13,626 particles, which presented the expected filament shape and D-loop features such as density attributable to the dsDNA arms of the target DNA. These particles were used for Topaz training and picking of 415,845 particles, of which 413,320 particles were extracted with a 208-pixel box. The extracted particles were fed directly into ab initio reconstruction and heterogeneous refinement, which yielded a 3D volume of a filament with recognisable D-loop features at lower map contouring for 328,724 extracted particles. 3D classification into 10 classes identified a single class of 43,824 particles of a complete D-loop structure, which refined in non-uniform refinement to 2.96 Å. This particle set was used in a second round of Topaz training and picking that yielded 922,204 particles, of which 900,213 were extracted with a 208-pixel box and used directly in ab initio reconstruction and heterogenous refinement of a filament volume of 711,480 particles with recognisable D-loop features at lower map contouring. 3D classification of this particle set over 10 classes identified one class of 74,483 particles corresponding to a high-quality D-loop structure, which was improved by further classification and refinement by heterogenous refinement. The resulting set of 102,679 particles was subject to non-uniform refinement to yield a 3D reconstruction at 2.72 Å, followed by reference-based motion correction of 102,596 particles and another step of non-uniform refinement, to yield a final reconstruction at 2.64 Å resolution.

## Model building and refinement

An atomic model comprising nine RAD51-ATP protomers, 26 nucleotides of ssDNA and 41 nucleotides for each strand of the 'mismatch bubble' dsDNA was built into the 3D volume of the final dataset #2 reconstruction at 2.64 Å resolution using Coot (*Emsley and Cowtan, 2004*). Fitting of the

model to the map was improved by real-space refinement in Phenix (*Afonine et al., 2018*) and model rebuilding in Coot (*Emsley and Cowtan, 2004*). In the later stages of refinement, map fitting and model stereochemistry were improved using ISOLDE (*Croll, 2018*). The register of the DNA sequence in the structure could be unambiguously assigned based on fitting of the nucleoside bases in the homologous dsDNA region, where the map resolution was highest.

## Optical tweezers-based single-molecule fluorescence imaging

To reconstitute RAD51 D-loop formation at the single-molecule level, we used a C-Trap (LUMICKS) instrument that combines optical tweezers and confocal fluorescence microscopy. The experimental setup featured a microfluidic chip with four distinct flow channels separated by laminar flow: one for streptavidin-coated polystyrene beads (Spherotech, 4.43 μm diameter), one for biotinylated $\lambda$ DNA, and two for buffer exchange or RAD51 nucleoprotein filaments reservoirs.

Doubly biotinylated $\lambda$ DNA molecules (48.5 kbp) prepared as previously described (*Candelli et al., 2014*) were tethered between two streptavidin-coated beads in the microfluidic flow cell. Successful capture of a single $\lambda$ DNA was confirmed by measuring force-extension curves (*Smith et al., 1996*). The optical traps were calibrated to a stiffness of 0.4 pN/nm, enabling stable manipulation of the DNA tether under tension (*Rouzina and Bloomfield, 2001a*; *Rouzina and Bloomfield, 2001b*). Fluorescence imaging was performed using a single 488 nm laser for excitation with an illumination intensity of 2 μW. Fluorescence signals were detected on a highly sensitive single-photon counting detector. Confocal fluorescence scans were performed over a region of 300 × 50 pixels to visualise the DNA tether and the edges of the beads. Pixel width was set to 0.1 μm, and each pixel was illuminated for 0.1 ms. During imaging, DNA was held at a constant force of 20 pN, and the flow channels were closed.

RAD51 D-loop formation reactions were examined by incubating 20 μM RAD51 with 1 μM Alexa 488-labelled ssDNA 49mer in buffer: 25 mM HEPES pH 7.5, 150 mM NaCl, 2 mM ATP, 2 mM CaCl$_2$, 2 mM DTT at room temperature for 20 min, 40-fold diluted in the same buffer and injected into the microfluidic flow cell for analysis of the interaction with the $\lambda$ DNA. The 49-nt-long sequence of the ssDNA is complementary to nucleotides 10,617–10,666 of $\lambda$ DNA, to promote D-loop formation at a single, specific site on the DNA tether. During imaging, the DNA tether tension was maintained at 20 pN, ensuring optimal conditions for detecting RAD51 interactions along the DNA substrate.

## Data extraction and analysis

Two-dimensional (2D) fluorescence scans (300×50 pixels) of individual $\lambda$ DNA molecules were recorded for a total acquisition time of 180 s at a temporal resolution of one frame every 2 s. Given that RAD51 binding induces DNA extension (*Hilario et al., 2009*; *van Mameren et al., 2009*), leading to drift of the D-loop position under the constant applied force regime, only the initial 30 frames were considered for analysis to ensure accurate localisation. The stack of frames was summed using the open-source image processing package FIJI (ImageJ), to enhance fluorescence signal detection by summing pixel intensity values across all frames in a stack (*Finkelstein et al., 2010*). Fluorescence intensity profiles were extracted by applying a linear scan along the DNA contour and bead peripheries, excluding regions near the bead peripheries to minimise fluorescence distortion, enabling precise quantification of binding events.

Due to fluorescence interference and reduced sensitivity near the beads, regions adjacent to each bead were excluded from analysis. The extent of this exclusion region was determined by performing a linear fit of the loss of fluorescence intensity measured at the junction areas versus the physical diameter of the beads and the contour length of $\lambda$ DNA. Based on this analysis, the exclusion region was estimated to be approximately 0.75 kbp from each end of the DNA molecule.

To automate peak identification, fluorescence intensity profiles were plotted against position along $\lambda$ DNA. Using the peak fitting module in Origin Pro (OriginLab), fluorescence intensity peaks were identified and fitted with a 2D-Gaussian model which tracks each peak using a symmetrical bell-shaped curve characterised by amplitude, width (full width at half maximum), and centre position (X-centre) to accurately locate and quantify binding events (*Morin et al., 2022*; *Gorman et al., 2007*). The X-centre values of the detected peaks were extracted to determine the precise binding positions along $\lambda$ DNA. Binding event frequencies were binned at 1 kbp intervals along the DNA molecule

length. Similar peak detection and Gaussian fitting were applied to the histogram of binding events, allowing for the identification of the focal binding site.

## Acknowledgements

We would like to thank Chen Qian for early work on the project and the Biochemistry cryoEM facility team for assistance with data collection. This work was funded by a Wellcome Trust Investigator award to LP (221892/Z/20/Z) and a UKRI BBSRC PhD studentship award to LP and RA (grant number 2271086).

## Additional information

### Funding

| Funder | Grant reference number | Author |
| --- | --- | --- |
| Wellcome Trust | 10.35802/221892 | Luca Pellegrini |
| Biotechnology and Biological Sciences Research Council | 2271086 | Robert E Appleby |

The funders had no role in study design, data collection and interpretation, or the decision to submit the work for publication. For the purpose of Open Access, the authors have applied a CC BY public copyright license to any Author Accepted Manuscript version arising from this submission.

### Author contributions

Luay Joudeh, Data curation, Supervision, Investigation, Methodology, Writing – review and editing; Robert E Appleby, Data curation, Funding acquisition, Investigation, Methodology, Writing – review and editing; Joseph D Maman, Conceptualization, Supervision, Methodology, Writing – review and editing; Luca Pellegrini, Conceptualization, Supervision, Funding acquisition, Writing – original draft, Writing – review and editing

### Author ORCIDs

Luay Joudeh ⓘ http://orcid.org/0000-0001-9338-205X
Luca Pellegrini ⓘ https://orcid.org/0000-0002-9300-497X

Reviewer #1 (Public review): https://doi.org/10.7554/eLife.107114.3.sa1
Reviewer #2 (Public review): https://doi.org/10.7554/eLife.107114.3.sa2
Reviewer #3 (Public review): https://doi.org/10.7554/eLife.107114.3.sa3
Author response https://doi.org/10.7554/eLife.107114.3.sa4

## Additional files

### Supplementary files

Supplementary file 1. Oligonucleotide sequence, size, and labelling.

Supplementary file 2. CryoEM data collection and real-space refinement.

MDAR checklist

### Data availability

Atomic coordinates and cryoEM maps for the human RAD51 D-loop have been deposited in the PDB and EMDB under accession codes 9I62 and EMD-52646, respectively.

The following datasets were generated:

| Author(s) | Year | Dataset title | Dataset URL | Database and Identifier |
|---|---|---|---|---|
| Pellegrini L, Joudeh L, Appleby RE | 2025 | CryoEM structure of a RAD51 D-loop | https://doi.org/10.2210/pdb9i62/pdb | Worldwide Protein Data Bank, 10.2210/pdb9i62/pdb |
| Pellegrini L, Joudeh L, Appleby RE | 2025 | CryoEM structure of a RAD51 D-loop | https://www.ebi.ac.uk/emdb/EMD-52646 | Electron Microscopy Data Bank, EMD-52646 |

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
