## [Editor Report · eLife Assessment]

This **landmark** study describes the structure of the human RAD51 filament with a recombination intermediate called the displacement loop (D-loop). Using cryogenic structural, biochemical, and single-molecule analyses, the authors provide **compelling** evidence on how the RAD51 filament promotes strand exchange between single-stranded and double-stranded DNAs. The findings are highly relevant to the fields of homologous recombination, DNA repair, and genome stability.

---

## [Referee Report · Reviewer #1 (Public review)]

Summary:

The paper describes the cryoEM structure of RAD51 filament on the recombination intermediate. In the RAD51 filament, the insertion of a DNA-binding loop called the L2 loop stabilizes the separation of the complementary strand for the base-pairing with an incoming ssDNA and the non-complementary strand, which is captured by the second DNA-binding channel called the site II. The molecular structure of the RAD51 filament with a recombination intermediate provides a new insight into the mechanism of homology search and strand exchange between ssDNA and dsDNA.

Strong points:

This is the first human RAD51 filament structure with a recombination intermediate called the D-loop. The work has been done with great care, and the results shown in the paper are compelling based on cryo-EM and biochemical analyses. The paper is really nice and important for researchers in the field of homologous recombination, which gives a new view on the molecular mechanism of RAD51-mediated homology search and strand exchange.

Comments on revisions:

The authors nicely address most of the previous points.

---

## [Referee Report · Reviewer #2 (Public review)]

Homologous recombination is essential for DNA double-strand break repair, with RAD51-catalyzed strand exchange at its core. This study presents a 2.64 Å resolution cryogenic electron microscopy structure of the RAD51 D-loop complex, achieved through reconstitution of a RAD51 mini-filament. The structure uncovers how specific RAD51 residues drive strand exchange, offering atomic-level insight into the mechanics of eukaryotic HR and DNA repair.

Comments on revisions:

Authors acknowledged:

"We acknowledge that there exists an extensive body of literature that has investigated the polarity of strand exchange by RecA and RAD51 under a variety of experimental conditions, and we have added a brief comment to the text to reflect this, as well as some of the key citations. Undoubtedly, and as we also mention in our reply to the public reviews, further experimental work will be needed for a full reconciliation of the available evidence."

In the revised manuscript, this is reflected in the statement:

"Our mechanistic interpretation of static D-loop structures awaits full reconciliation with earlier efforts to determine strand-exchange polarity for RecA and RAD51 measured under a variety of experimental conditions."

Among the four cited studies, my understanding (as a person who has never studied this subject of polarity) is as follows:

•References 50 (EMBO J. 1997), 51 (Cell. 1995), and 52 (Nature. 2008) suggest that the strand exchange by human RAD51 occurs with a polarity opposite to that of RecA-that is, in the 5′→3′ direction relative to the complementary strand, or 3′→5′ relative to the initiating single-stranded DNA (isDNA).

• In contrast, reference 49 (PNAS 1998) proposed that 5′→3′ polarity (relative to isDNA) is conserved across RecA, human RAD51, and yeast RAD51.

Given the substantial structural analysis provided in the current manuscript, it would strengthen the work to include a concise description of these earlier biochemical findings, rather than citing them without context. This would benefit readers who are not familiar with the longstanding studies in the field and allow for a more informed interpretation of how the structural observations may reconcile or contrast with previous work.

---

## [Referee Report · Reviewer #3 (Public review)]

Summary:

Built on their previous pioneer expertise in studying RAD51 biology, in this paper, the authors aim to capture and investigate the structural mechanism of human RAD51 filament bound with a displacement loop (D-loop), which occurs during the dynamic synaptic state of the homologous recombination (HR) strand-exchange step. As the structures of both pre- and post-synaptic RAD51 filaments were previously determined, a complex structure of RAD51 filament during strand exchange is one of the key missing pieces of information for a complete understanding of how RAD51 functions in HR pathway. This paper aims to determine the high-resolution cryo-EM structure of RAD51 filament bound with D-loop. Combined with mutagenesis analysis and biophysical assays, the authors aim to investigate the D-loop DNA structure, RAD51 mediated strand separation and polarity, and a working model of RAD51 during HR strand invasion in comparison with RecA.

Strengths:

(1) The structural work and associated biophysical assays in this paper are solid, elegantly designed and interpreted. These results provide novel insights into RAD51's function in HR.

(2) The DNA substrate used was well designed, taking into consideration of the nucleotide number requirement of RAD51 for stable capture of donor DNA. This DNA substrate choice lays the foundation for successfully determining the structure of the RAD51 filament on D-loop DNA using single-partial cryo-EM.

(3) The authors utilised their previous expertise in capping DNA ends using monometric streptavidin and combined their careful data collection and processing to determine the cryo-EM structure of full-length human RAD51 bound at D-loop in high resolution. This interesting structure forms the core part of this work and allows detailed mapping of DNA-DNA and DNA-protein interaction among RAD51, invading strands, and donor DNA arms (Figures 1, 2, 3, 4). The geometric analysis of D-loop DNA bound with RAD51 and EM density for homologous DNA pairing are also impressive (Figure S5). The previously disordered RAD51's L2-loop is now ordered and traceable in the density map and functions as a physical spacer when bound with D-loop DNA. Interestingly, the authors identified that the side chain position of F279 in the L2_loop of RAD51_H differs from other F279 residues in L2-loops of E, F and G protomers. This asymmetric binding of L2 loops and RAD51_NTD binding with donor DNA arms forms the basis of the proposed working model about the polarity on csDNA during RAD51-mediated strand exchange.

(4) This work also includes mutagenesis analysis and biophysical experiments, especially EMSA, single-molecule fluorescence imaging using an optical tweezer, and DNA strand exchange assay, which are all suitable methods to study the key residues of RAD51 for strand exchange and D-loop formation (Figure 5).

Weaknesses:

(1) The proposed model for the 3'-5' polarity of RAD51-mediated strand invasion is based on the structural observations in the cryo-EM structure. This study lacks follow-up biochemical/biophysical experiments to validate the proposed model compared to RecA or developing methods to capture structures of any intermediate states with different polarity models.

(2) The functional impact of key mutants designed based on structure has not been tested in cells to evaluate how these mutants impact the HR pathway.

The significance of the work for the DNA repair field and beyond:

Homologous recombination (HR) is a key pathway for repairing DNA double-strand breaks and involves multiple steps. RAD51 forms nucleoprotein filaments first with 3' overhang single-strand DNA (ssDNA), followed by a search and exchange with a homology strand. This function serves as the basis of an accurate template-based DNA repair during HR. This research addressed a long-standing challenge of capturing RAD51 bound with the dynamic synaptic DNA and provided the first structural insight into how RAD51 performs this function. The significance of this work extends beyond the discovery biology for the DNA repair field, into its medical relevance. RAD51 is a potential drug target for inhibiting DNA repair in cancer cells to overcome drug resistance. This work offers a structural understanding of RAD51's function with D-loop and provides new strategies for targeting RAD51 to improve cancer therapies.

---

## [Author Response]

The following is the authors’ response to the original reviews.

**Reviewer #1 (Public review):**
Summary:The paper describes the cryoEM structure of RAD51 filament on the recombination intermediate. In the RAD51 filament, the insertion of a DNA-binding loop called the L2 loop stabilizes the separation of the complementary strand for the base-pairing with an incoming ssDNA and the non-complementary strand, which is captured by the second DNA-binding channel called the site II. The molecular structure of the RAD51 filament with a recombination intermediate provides a new insight into the mechanism of homology search and strand exchange between ssDNA and dsDNA.Strengths:This is the first human RAD51 filament structure with a recombination intermediate called the D-loop. The work has been done with great care, and the results shown in the paper are compelling based on cryo-EM and biochemical analyses. The paper is really nice and important for researchers in the field of homologous recombination, which gives a new view on the molecular mechanism of RAD51-mediated homology search and strand exchange.Weaknesses:The authors need more careful text writing. Without page and line numbers, it is hard to give comments.

We would like to thank the reviewer for their kind words of appreciation of our work.

**Reviewer #2 (Public review):**
Summary:Homologous recombination (HR) is a critical pathway for repairing double-strand DNA breaks and ensuring genomic stability. At the core of HR is the RAD51-mediated strand-exchange process, in which the RAD51-ssDNA filament binds to homologous double-stranded DNA (dsDNA) to form a characteristic D-loop structure. While decades of biochemical, genetic, and single-molecule studies have elucidated many aspects of this mechanism, the atomic-level details of the strand-exchange process remained unresolved due to a lack of atomic-resolution structure of RAD51 D-loop complex.In this study, the authors achieved this by reconstituting a RAD51 mini-filament, allowing them to solve the RAD51 D-loop complex at 2.64 Å resolution using a single particle approach. The atomic resolution structure reveals how specific residues of RAD51 facilitate the strand exchange reaction. Ultimately, this work provides unprecedented structural insight into the eukaryotic HR process and deepens the understanding of RAD51 function at the atomic level, advancing the broader knowledge of DNA repair mechanisms.Strengths:The authors overcame the challenge of RAD51's helical symmetry by designing a minifilament system suitable for single-particle cryo-EM, enabling them to resolve the RAD51 D-loop structure at 2.64 Å without imposed symmetry. This high resolution revealed precise roles of key residues, including F279 in Loop 2, which facilitates strand separation, and basic residues on site II that capture the displaced strand. Their findings were supported by mutagenesis, strand exchange assays, and single-molecule analysis, providing strong validation of the structural insights.Weaknesses:Despite the detailed structural data, some structure-based mutagenesis data interpretation lacks clarity. Additionally, the proposed 3′-to-5′ polarity of strand exchange relies on assumptions from static structural features, such as stronger binding of the 5′-arm-which are not directly supported by other experiments. This makes the directional model compelling but contradicts several well-established biochemical studies that support a 5'-to-3' polarity relative to the complementary strand (e.g., Cell 1995, PMID: 7634335; JBC 1996, PMID: 8910403; Nature 2008, PMID: 18256600).Overall:The 2.6 Å resolution cryoEM structure of the RAD51 D-loop complex provides remarkably detailed insights into the residues involved in D-loop formation. The high-quality cryoEM density enables precise placement of each nucleotide, which is essential for interpreting the molecular interactions between RAD51 and DNA. Particularly, the structural analysis highlights specific roles for key domains, such as the N-terminal domain (NTD), in engaging the donor DNA duplex.This structural interpretation is further substantiated by single-molecule fluorescence experiments using the KK39,40AA NTD mutant. The data clearly show a significant reduction in D-loop formation by the mutant compared to wild-type, supporting the proposed functional role of the NTD observed in the cryoEM model.However, the strand exchange activity interpretation presented in Figure 5B could benefit from a more rigorous experimental design. The current assay measures an increase in fluorescence intensity, which depends heavily on the formation of RAD51-ssDNA filaments. As shown in Figure S6A, several mutants exhibit reduced ability to form such filaments, which could confound the interpretation of strand exchange efficiency. To address this, the assay should either: (1) normalize for equivalent levels of RAD51-ssDNA filaments across samples, or (2) compare the initial rates of fluorescence increase (i.e., the slope of the reaction curve), rather than endpoint fluorescence, to better isolate the strand exchange activity itself.

We agree with the reviewer that the reduced filament-forming ability of some of the RAD51 mutants complicates a straightforward interpretation of their strand-exchange assay. Interestingly, the RAD51 mutants that appear most impaired are the esDNA-capture mutants that do not contact the ssDNA in the structure of the pre-synaptic filament. However, the RAD51 NTD mutants, that display the most severe defect in strand-exchange, have a near-WT filament forming ability.

Based on the structural features of the D-loop, the authors propose that strand pairing and exchange initiate at the 3'-end of the complementary strand in the donor DNA and proceed with a 3'-to-5' polarity. This conclusion, drawn from static structural observations, contrasts with several well-established biochemical studies that support a 5'-to-3' polarity relative to the complementary strand (e.g., Cell 1995, PMID: 7634335; JBC 1996, PMID: 8910403; Nature 2008, PMID: 18256600). While the structural model is compelling and methodologically robust, this discrepancy underscores the need for further experiments.

We would like to thank the reviewer for highlighting the importance of our findings to our understanding of the mechanism of homologous recombination.

The reviewer correctly points out that the polarity of strand exchange by RecA and RAD51 is an extensively researched topic that has been characterised in several authoritative studies. In our paper, we simply describe the mechanistic insights obtained from the structural D-loop models of RAD51 (our work) and RecA (Yang et al, PMID: 33057191).The structures illustrate a very similar mechanism of Dloop formation that proceeds with opposite polarity of strand exchange for RAD51 and RecA. Comparison of the D-loop structures for RecA and RAD51 provides an attractive explanation for the opposite polarity, as caused by the different positions of their dsDNA-binding domains in the filament structure.

We agree with the reviewer that further investigation will be needed for an adequate rationalisation of the available evidence. We will mention the relevant literature in the revised version of the manuscript.

**Reviewer #3 (Public review):**
Summary:Built on their previous pioneer expertise in studying RAD51 biology, in this paper, the authors aim to capture and investigate the structural mechanism of human RAD51 filament bound with a displacement loop (D-loop), which occurs during the dynamic synaptic state of the homologous recombination (HR) strand-exchange step. As the structures of both pre- and post-synaptic RAD51 filaments were previously determined, a complex structure of RAD51 filaments during strand exchange is one of the key missing pieces of information for a complete understanding of how RAD51 functions in the HR pathway. This paper aims to determine the high-resolution cryo-EM structure of RAD51 filament bound with the D-loop. Combined with mutagenesis analysis and biophysical assays, the authors aim to investigate the D-loop DNA structure, RAD51-mediated strand separation and polarity, and a working model of RAD51 during HR strand invasion in comparison with RecA.Strengths:(1) The structural work and associated biophysical assays in this paper are solid, elegantly designed, and interpreted. These results provide novel insights into RAD51's function in HR.(2) The DNA substrate used was well designed, taking into consideration the nucleotide number requirement of RAD51 for stable capture of donor DNA. This DNA substrate choice lays the foundation for successfully determining the structure of the RAD51 filament on D-loop DNA using single-particle cryo-EM.(3) The authors utilised their previous expertise in capping DNA ends using monomeric streptavidin and combined their careful data collection and processing to determine the cryo-EM structure of full-length human RAD51 bound at the D-loop in high resolution. This interesting structure forms the core part of this work and allows detailed mapping of DNA-DNA and DNA-protein interaction among RAD51, invading strands, and donor DNA arms (Figures 1, 2, 3, 4). The geometric analysis of D-loop DNA bound with RAD51 and EM density for homologous DNA pairing is also impressive (Figure S5). The previously disordered RAD51's L2-loop is now ordered and traceable in the density map and functions as a physical spacer when bound with D-loop DNA. Interestingly, the authors identified that the side chain position of F279 in the L2_loop of RAD51_H differs from other F279 residues in L2-loops of E, F, and G protomers. This asymmetric binding of L2 loops and RAD51_NTD binding with donor DNA arms forms the basis of the proposed working model about the polarity of csDNA during RAD51-mediated strand exchange.(4) This work also includes mutagenesis analysis and biophysical experiments, especially EMSA, singlemolecule fluorescence imaging using an optical tweezer, and DNA strand exchange assay, which are all suitable methods to study the key residues of RAD51 for strand exchange and D-loop formation (Figure 5).Weaknesses:(1) The proposed model for the 3'-5' polarity of RAD51-mediated strand invasion is based on the structural observations in the cryo-EM structure. This study lacks follow-up biochemical/biophysical experiments to validate the proposed model compared to RecA or developing methods to capture structures of any intermediate states with different polarity models.(2) The functional impact of key mutants designed based on structure has not been tested in cells to evaluate how these mutants impact the HR pathway.The significance of the work for the DNA repair field and beyond:Homologous recombination (HR) is a key pathway for repairing DNA double-strand breaks and involves multiple steps. RAD51 forms nucleoprotein filaments first with 3' overhang single-strand DNA (ssDNA), followed by a search and exchange with a homologous strand. This function serves as the basis of an accurate template-based DNA repair during HR. This research addressed a long-standing challenge of capturing RAD51 bound with the dynamic synaptic DNA and provided the first structural insight into how RAD51 performs this function. The significance of this work extends beyond the discovery of biology for the DNA repair field, into its medical relevance. RAD51 is a potential drug target for inhibiting DNA repair in cancer cells to overcome drug resistance. This work offers a structural understanding of RAD51's function with the D-loop and provides new strategies for targeting RAD51 to improve cancer therapies.

We thank the reviewer for their positive comments on the significance of our work. Concerning the proposed polarity of strand exchange based on our structural finding, please see our reply to the previous reviewer; we agree with the reviewer that further experimentation will be needed to to reach a settled view on this.

Testing the functional effects of the RAD51 mutants on HR in cells was not an aim of the current work but we agree that it would be a very interesting experiment, which would likely provide further important insights into the mechanism of strand exchange at the core of the HR reaction.

**Reviewer #1 (Recommendations for the authors):**
Major points:(1) Structural analysis showed a critical role of F279 in the L2 loop. However, the biochemical study showed that the F279A substitution did not provide a strong defect in the in vitro strand exchange, as shown in Figure 5B. Moreover, a previous study by Matsuo et al. FEBS J, 2006; ref 43 showed human RAD51-F279A is proficient in the in vitro strand exchange. These suggest that human RAD51 F279 is not critical for the strand exchange. The authors need more discussions of the role of F279 or the L2 for the RAD51-mediated reactions in the Discussion.

In the strand-exchange essay of Figure 5B, the F279A mutant shows the mildest phenotype, in agreement with the findings of Matsuo et al. Accordingly, in the text we describe the F279A mutant as having a “modest impact” on strand-exchange.

We have now added a brief comment to the relevant text, pointing out that the result of the strand exchange assay for F279A are in agreement with the previous findings by Matsuo et al., and adding the reference.

(2) In some parts, the authors cited the newest references rather than the paper describing the original findings. For RAD51 paralogs, why are these three (refs 21,22, 23) selected here? For FIGNL1, why is only one (ref 24) chosen?

The cited publications were chosen to acquaint the reader with the latest structural and mechanistic advances about the function of some of the most important and well-studied recombination mediator proteins. For completeness, we have now added a further reference for FIGNL1 - Ito, Masaru et al, Nat Comm, 2023 – in the Introduction, to provide the reader with an additional pointer to our current knowledge about the mechanism of FIGNL1 in Homologous Recombination.

Minor points:(1) Page 3, line 1 in the second paragraph, the reaction of "HR": HR should be homology search and strand exchange. HR is used incorrectly throughout the text, please check them. Remove "strandexchange" from ATPases in line 2.

We believe that HR is used correctly in this context, as we refer to the biochemical reactions of HR, which includes the search for homology and strand exchange.

We have removed “strand-exchange” from ATPases in line 2, as requested by the reviewer.

(2) Supplementary Figure 1B, C, "EMSA" experiment: Please indicate an experimental condition in the legend: how ssDNA and dsDNA were mixed with RAD51. In (B), this is not an actual EMSA result, but rather a native gel analysis of reaction products with the D-loop. In (C), was the binding of RAD51 to the pre-formed D-loop examined? Which is correct here? Moreover, why do the authors need streptavidin in this experiment? Please explain why this is necessary for the EMSA assay. Please show where is Cy3 or Cy5 labels on the DNAs should be shown in the schematic drawing.

The conditions for the experiments of Supplementary figure 1B, C are reported in the Methods section.

Panel B shows the mobility shifts of the ssDNA and dsDNA sequences in panel A, so it is appropriate to describe it as an EMSA.

We did not examine the binding of RAD51 to a pre-formed D-loop.

We used streptavidine in the experiment of Supplementary Figure 1C to show that streptavidine binding did not interfere with D-loop reconstitution.

The position of the Cy3, Cy5 labels in the DNAs is reported in Table S1.

(3) Figure S4B, page 6, line 6 from the top, 5'-arm and 3'-arm: please add them to the figure. And also, please explain what 5'-arm and 3'-arm are here in the text, as shown in lines 3-5 in the second paragraph of the same page.

We thank the reviewer for spotting this slight incongruity. We have removed the reference to 5’- and 3’arms of the donor DNA in the initial description of the D-loop (first paragraph of the “D-loop structure” section, 6 lines from the top), as the nomenclature for the arms of the donor DNA is introduced more appropriately in the following paragraph. Thus, there is no need to re-label Figure S4B; we note that the 5’- and 3’-labels are added to the arms of the donor DNA in Figure S4D.

(4) Page 7, line 4, and Figure 2E, "C24": C24 should be C26 here (Figure 2D shows that position 24 in esDNA is "T").

We thank the reviewer for spotting this typo, that is now corrected in the revised version of Figure 2 and in the text.

(5) Page 8, line 1, K284: It would be nice to show "K284" in Figure 3F.

We have added the side chain of K284 to Figure 3F, as suggested by the reviewer.

(6) Page 8, second paragraph, line 3 from the bottom, "5'-arm" should be "3'-arm" for the binding of RAD51A NTD to ds DNA (Figure 4D).

We thank the reviewer for spotting this typo, that is now corrected in the revised version of the text.

**Reviewer #2 (Recommendations for the authors):**
I understand that the strand exchange polarity of RAD51 should be opposite to that of RecA. But in the RecA manuscript (Nature 2020), it states (in the extended figure 1) " Because the mini-filament consists of fused RecA protomers, it does not reflect the effects a preferential polarity of RecA polymerization might have on the directionality of strand exchange. Also, our strand exchange reactions do not include the single-stranded DNA binding protein SSB that is involved in strand exchange in vivo and may sequester released DNA strands."

We are aware that the findings by Yang et al, 2020 were obtained with a multi-protomeric RecA chimera and that their construct might not therefore recapitulate a potential effect of RecA polymerisation on the directionality of strand-exchange.

Comparison of the RecA and RAD51 D-loop structures shows that RecA and RAD51 adopt the same asymmetric mechanism of D-loop formation, which begins at one arm of the donor DNA and proceeds with donor unwinding and strand invasion until the second arm is captured, completing D-loop formation. However, the cryoEM structures provide compelling evidence that, after engagement with the donor DNA, RecA and RAD51 proceed to unwind the donor with opposite polarity; the structures provide a clear rationale for this, because of the different position of their dsDNA-binding domains relative to the ATPase domain.

We acknowledge that there exists an extensive body of literature that has investigated the polarity of strand exchange by RecA and RAD51 under a variety of experimental conditions, and we have added a brief comment to the text to reflect this, as well as some of the key citations. Undoubtedly, and as we also mention in our reply to the public reviews, further experimental work will be needed for a full reconciliation of the available evidence.

**Reviewer #3 (Recommendations for the authors):**
(1) I have a minor comment regarding the DNA shown in the structural figures in this work. The authors have used different colours to differentiate between isDNA, esDNA, and csDNA for easier interpretation. However, these colour codes are inconsistent across Figures 1, 2, 3, and 5. This inconsistency makes it difficult to interpret which strand is which, particularly for readers unfamiliar with D-loops and strand invasion. A consistent colour scheme for the DNA strands would enhance the quality of the structural figures.

We appreciate the reviewer’s comment about the colour scheme of the strands in the D-loop. We chose a unique colour scheme for each figure, to help the reader focus on the particular structural features that we wanted to highlight in the figure. So for instance, in figure 1D we chose to highlight the relationship (complementary vs identical) of the donor DNA strands with the the invading strand; in figure 2, the emphasis is on distinguishing the homologously paired dsDNA (pink) from the exchanged strand (magenta), as a consequence of L2 loop binding; etc.

(2) I have another comment regarding the rationale behind naming the RAD51 protomers (A to H) within the structure, which could confuse general readers if not clearly explained. In this paper, the RAD51 protomer is RAD51_A when closest to the 3' end of the isDNA. I assume the authors chose this order because HR generates a 3' ssDNA overhang before strand invasion. It would be beneficial for the introduction and results sections to mention this property of the 3' ssDNA overhang and the reasoning behind this naming strategy. This explanation will help readers understand how it differs from other naming orders used in RecA/RAD51 with ssDNA, where protomer A is closer to the 5' ssDNA.

We thank the reviewer for their insightful comment. We chose to name as chain A the RAD51 protomer nearest to the 3’-end of the isDNA to be consistent with the naming scheme that we use for all our published RAD51 filament structures.

(3) I have highlighted some text within this paper that has contradicting parts for authors to clarify and correct:"Overall, the structural features of the RAD51 D-loop provide a strong indication that strand pairing and exchange begins at the 3'-end of the complementary strand in the donor DNA and progresses with 3'-to5' polarity (Fig. 5F)""The observed 5'-to-3' polarity of strand-exchange by RAD51 is opposite to the 3'-to-5' polarity of bacterial RecA (Fig. S8), that was determined based on cryoEM structures of RecA D-loops".

We thank the reviewer for alerting us to this inconsistency that has now been corrected in the revised manuscript.

(4) Figure S8 last model: NTD should be CTD in the title; Figure 2B: resolution scale bar needs A unit. We thank the reviewer for spotting this typo that has now been corrected in the revised version of figure S8.

We couldn’t find a missing resolution scale bar in Figure 2B; however, we have added a missing resolution bar with A unit to Fig. S3B.